# Overview of Addressed Fiber Bragg Structures' Development

Timur Agliullin [1], German Il'In [2], Artem Kuznetsov [1], Rinat Misbakhov [1], Rustam Misbakhov [3], Gennady Morozov [1], Oleg Morozov [1,*], Ilnur Nureev [1] and Airat Sakhabutdinov [1]

1   Department of Radiophotonics and Microwave Technologies, Kazan National Research Technical University Named after A.N. Tupolev-KAI, 10 K. Marx St., Kazan 420111, Russia
2   Department of Electronic and Quantum Means of Information Communication, Kazan National Research Technical University Named after A.N. Tupolev-KAI, 10 K. Marx St., Kazan 420111, Russia
3   Almetyevsk Branch, Kazan National Research Technical University Named after A.N. Tupolev-KAI, 9b Stroiteli Avenue, Almetyevsk 423400, Russia
*   Correspondence: ogmorozov@kai.ru

**Abstract:** An addressed fiber Bragg structure (AFBS) is a special type of fiber Bragg grating simultaneously performing the functions of a two-frequency radiation shaper and a sensitive element. An AFBS forms a two-frequency optical spectral response at its output, the difference frequency of which is invariant to measured physical fields and is referred to as the address frequency of the AFBS. Each of the AFBSs in the system has its own address frequency; therefore, a number of such structures can be interrogated simultaneously enabling the addressed multiplexing. In this article, we provide an overview of the theory and technology of AFBS, including the structures with three or more spectral components with various combinations of difference frequencies, both symmetrical and asymmetric. The subjects of interrogation of AFBSs, their fabrication and calibration are discussed as well. We also consider a wide range of applications in which AFBS can be used, covering such areas as oil and gas production, power engineering, transport, medicine, etc. In addition, the prospects for the further development of AFBS are proposed that mitigate the shortcomings of the current AFBSs' state of the art and open up new possibilities of their application.

**Keywords:** addressed fiber Bragg structure; interrogation of addressed fiber Bragg structures; fabrication of addressed fiber Bragg structures; fiber Bragg grating sensor implementation; microwave photonics

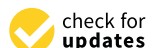



## 1. Introduction

Fiber Bragg gratings (FBG) have acquired widespread application since their introduction in the late 1970s [1], especially as photonic sensing elements for the measurement of various physical fields. Their attractive properties, such as small footprint and low weight, immunity to electromagnetic disturbances, high sensitivity and possibility of multiplexing several FBGs into a single system, provide advantages in numerous areas, including aerospace [2], automotive [3], biomedical [4], civil engineering, oil and gas industries [5], etc.

Several techniques for FBG multiplexing and interrogation have been developed to date. The most common approaches, such as wavelength [6], frequency [7], time [8] and spatial [9] division multiplexing, are implemented using complex and costly optoelectronic devices, such as spectrum analyzers, tunable Fabry–Perot interferometers, diffraction gratings, etc. Another problem of the traditional interrogating methods is the lack of sensor addressability, which leads to interrogation errors when spectrum overlapping takes place. In order to mitigate this issue, optical spectrum-coded FBG interrogation methods were proposed [10,11], in which the sensors are interrogated in real time according to autocorrelation between the sensor spectra and its code signature, thus allowing several FBGs to be distinguished within the same spectral range.

A different approach was proposed in which FBG performs a triple function: besides sensing, it acts as a two-frequency radiation shaper as well as enables address-based

multiplexing. Such FBGs are referred to as addressed fiber Bragg structures (AFBS) [12,13]. An AFBS is a type of FBG, the spectral response of which has two narrow components (notches). When an AFBS is connected to a wideband optical source, it forms an output radiation consisting of two narrowband frequencies, the difference between which is called the address frequency and belongs to the microwave range (GHz). The address frequency is invariant to the AFBS central wavelength shift when it is exposed to strain or temperature variations. Therefore, the address frequency is used as a distinguishing parameter, which makes it possible to interrogate several AFBSs even if their central wavelengths coincide.

The usage of AFBS significantly simplifies the interrogation scheme compared with the abovementioned optoelectronic methods, as it requires only a broadband light source, an optical filter with a predefined frequency response with an inclined profile and a photodetector.

The concept of AFBS was subsequently expanded to include the structures with three or more spectral components forming two or more address frequencies, which are also known as multi-addressed fiber Bragg structures (MAFBS) [14]. The increased number of address frequencies allows the enhancement of the accuracy of the central wavelength determination as well as the expansion of the sensor capacity of the system.

The current paper presents a comprehensive classification of addressed fiber Bragg structures, including both AFBSs and MAFBSs with various relative positions of the spectral address components. The theoretical and technological aspects of AFBS implementation are also discussed. An overview of a wide range of AFBS applications is given along with the directions of further AFBS development.

## 2. Classifications of Addressed Fiber Bragg Structures

The following classifications consider AFBSs with up to three spectral components. AFBSs with four or more address components can be classified in the same manner. Two approaches to the formation of AFBS have been proposed: the introduction of two or more phase $\pi$-shifts into the periodic structure of FBG ($N\pi$-FBG, where $N$ is the number of phase shifts, Figure 1) [15] and the sequential recording of several ultra-narrowband FBGs with different central wavelengths ($N\lambda$-FBG, Figure 2) [16]. For the former type, the transmitted radiation is used for interrogation, while for the latter type, the reflected light is utilized. Thus, the reflecting and transmitting AFBSs constitute the first AFBS classification.

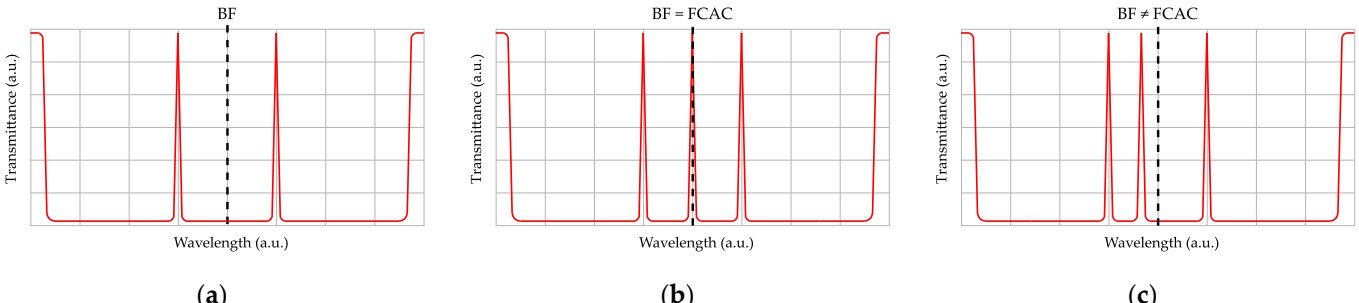

**Figure 1.** Transmitted spectrum of AFBS ($N\pi$-FBG type): (**a**) double-component AFBS (DCAFBS) with the Bragg frequency (BF) in between the frequencies of lateral address components (FLACs); (**b**) symmetric triple-component AFBS (STCAFBS) with the BF coinciding with the frequency of the central address component (FCAC); (**c**) asymmetric triple-component AFBS (ATCAFBS) with the BF not coinciding with the FCAC.

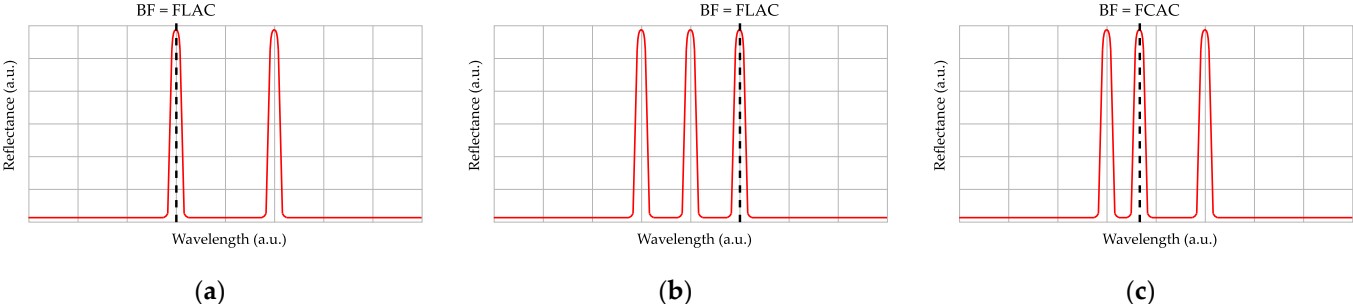

**Figure 2.** Reflected spectrum of AFBS (*N*λ-FBG type) with: (**a**) double-component AFBS (DCAFBS) with the Bragg frequency (BF) coinciding with the left frequency of lateral address component (FLAC); (**b**) symmetric triple-component AFBS (STCAFBS) with the BF coinciding with the right FLAC; (**c**) asymmetric triple-component AFBS (ATCAFBS) with the BF coinciding with the frequency of the central address component (FCAC).

The second classification of AFBS is according to the number of address frequencies: a single-addressed AFBS (Figures 1a and 2a), which is a double-component AFBS (DCAFBS), and a two-addressed AFBS (Figures 1b and 2b), which is a symmetrical triple-component AFBS (STCAFBS) with lateral address components spaced at the same address frequency from the central address component. A three-addressed AFBS (Figures 1c and 2c) is an asymmetric triple-component AFBS (ATCAFBS) with lateral address components spaced at different address frequencies from the central address component.

The third classification is the classification according to the coincidence of the frequency of the central address component and the Bragg frequency of the entire structure as a whole. The possibility of Bragg frequency definition for the entire structure as a whole follows from the invariance of the position of the address components when physical fields are applied to the structure. The Bragg frequency can be defined to be in the middle between the frequencies of lateral address components (FLACs) (Figure 1a) coinciding with the frequency of the central address component (FCAC), as in the case of STCAFBS (Figure 1b), or it cannot coincide with FCAC, as in the case of ATCAFBS (Figure 1c).

Alternatively, the Bragg wavelength can be defined to be coinciding with the FLAC of the DCAFBS (Figure 2a), or the STCAFBS (Figure 2b), or to be coinciding with the FCAC of the ATCAFBS (Figure 2c).

## 3. Interrogation of Addressed Fiber Bragg Structures

Typical schemes for AFBS interrogation are presented in Figure 3. The scheme for the transmitting *N*π-FBG AFBS is shown in Figure 3A, while Figure 3B represents the one for the reflecting *N*λ-FBG structures. In the schemes below, double-component AFBSs are used as an example. The schemes for the structures with three or more components are designed in the same way.

The schemes operate as follows. An optical source (1) generates a wideband optical radiation (insertion a), the bandwidth of which covers the whole range of AFBS components' wavelength shifts. The radiation passes through N addressed structures 2.1–2.N connected either in parallel using fiber-optic splitter nine and combiner ten (in the case of transmitting AFBS, Figure 3A), or sequentially (in the case of reflecting AFBS, Figure 3B). At the output of each AFBS, a radiation with two spectral components is formed, the spacing between which corresponds to the address frequency of the AFBS and is unique for each sensor in the system. The combined multi-frequency radiation from all the AFBSs (insertions b and c) is divided by means of a fiber-optic splitter six into two measuring and one reference channels. In each measuring channel, the radiation passes through an optical filter (3.1 and 3.2) with a linear inclined frequency response, which modifies the amplitudes of the frequency components of the AFBSs according to its known frequency response. The key difference between the measuring channels is that the optical filters 3.1 and 3.2 have different known temperature sensitivities of their spectral responses. The optical radiation

in all the channels is received by the corresponding photodetectors 4.1, 4.2 and 7, at the output of which the electric beating signals are generated at the address frequencies of the AFBSs. The electric signals are digitized using ADCs 5.1, 5.2 and 8, and the subsequent calculations of the AFBS central wavelengths are carried out for the ratio of the power of each measuring channel to the power in the reference channel. This eliminates the influence of the optical source fluctuations on the AFBS interrogation process.

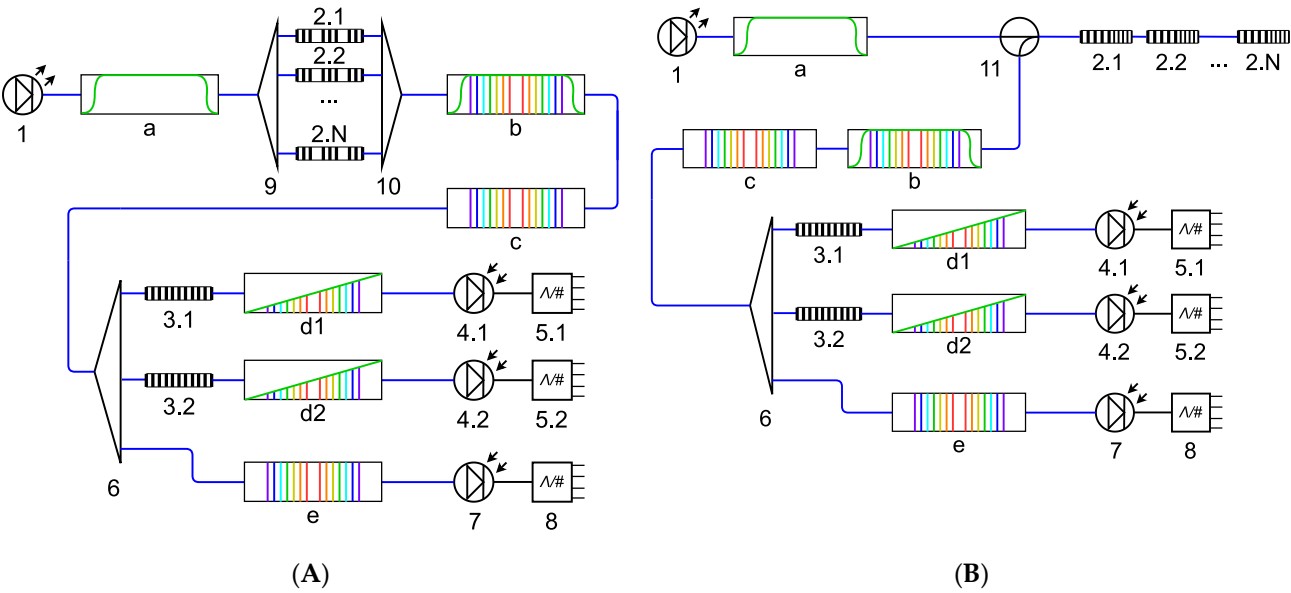

(**A**)    (**B**)

**Figure 3.** Interrogation schemes for AFBSs of: (**A**) transmitting type; (**B**) reflecting type; (1) wideband light source, (2.1)–(2.N) AFBS sensors, (3.1) and (3.2) optical filters with linear inclined frequency responses, (4.1) and (4.2) photodetectors of the measuring channels, (5.1) and (5.2) ADCs of measuring channels, (6) and (9) fiber-optic splitters, (7) photodetector of the reference channel, (8) ADC of the reference channel, (10) fiber-optic combiner, (11) fiber-optic circulator; inserted diagrams: (a) spectrum of the wideband light source, (b) and (c) spectra of light propagated through the AFBS sensors, (d1) and (d2) spectra of AFBS sensors at the output of the optical filter, (e) spectra of AFBSs in the reference channel; blue connection lines represent optical fibers, black connection lines represent electrical wires.

One of the key components of the AFBS interrogation scheme is the optical filter with a linear inclined frequency response. Such filters can be fabricated based on an FBG with the known spectral response having linear slopes. The deviation from the linear approximation of the optical filter frequency response is one of the main components of the measurement error, since the position of the AFBS spectrum is determined relative to it. Therefore, for the given FBG-based filter, the acceptable range of the AFBS spectral components shifts is determined, in which the deviation of the filter spectrum from the linear approximation does not exceed the desired value, as it is discussed by the authors in [17].

Figure 4 illustrates the principle of AFBS interrogation. Colored lines in Figure 4a denote the different variants of spectral positions of the two AFBSs relative to the spectral response of the optical filter, and in Figure 4b the corresponding RF spectra at the photodetector output are indicated in the same colors.

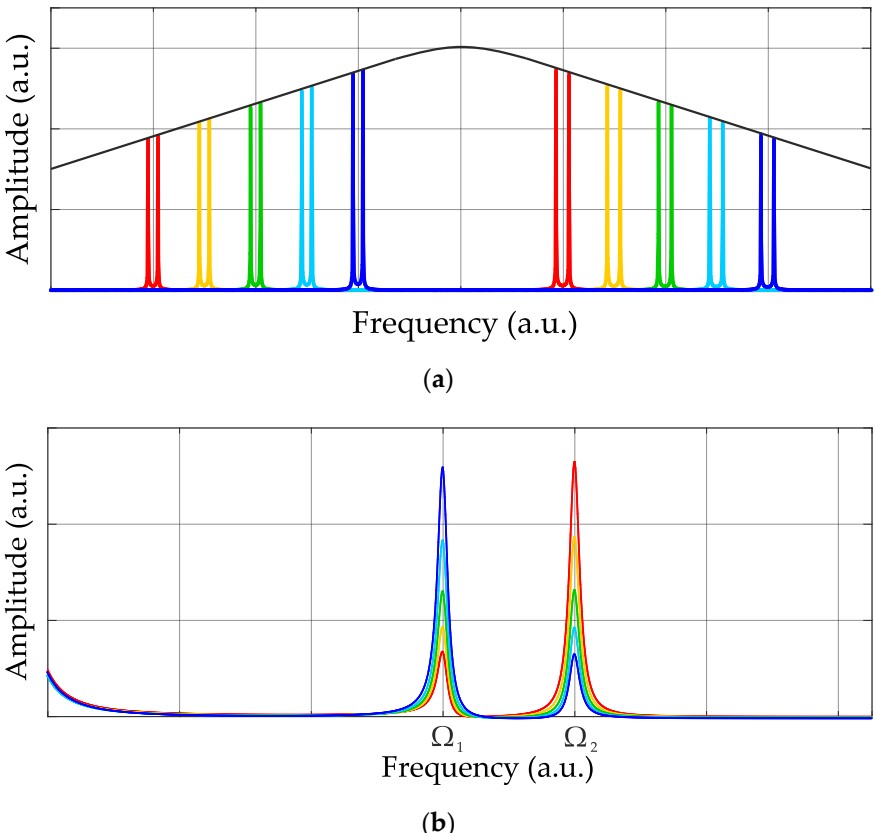

**Figure 4.** Interrogation of two double-component AFBSs with the address frequencies $\Omega_1$ and $\Omega_2$, respectively, with different variants of Bragg frequency: (**a**) spectral positions of AFBSs (colored lines) relative to the spectral response of the optical filter (black line); (**b**) corresponding spectra of the beating signal at the output of the photodetector.

As it follows from the principle of AFBS interrogation, in order to correctly determine the AFBS central wavelength, it is necessary to take into account the temperature drifts of the optical filters 3.1 and 3.2. For this reason, the filters are located close to each other in the system layout so that their temperature is assumed to be the same. Therefore, knowing the difference between the center wavelengths of the same AFBS determined using the filters, it is possible to calculate their temperature using the pre-defined temperature characteristics of the filters [17]. After that, the estimated value of temperature is used to calculate the absolute value of the center frequency of the filter, based on which the correction to the center frequency of the AFBS is determined.

Consider the output optical radiation of the *i*-th AFBS, which is represented as a sum of two harmonic oscillations spaced by the address frequency $\Omega_i$:

$$E_i(t) = A_i e^{j\omega_i t + \varphi_{Ai}} + B_i e^{j(\omega_i + \Omega_i)t + \varphi_{Bi}}, \tag{1}$$

where $A_i$ and $B_i$ are the amplitudes of the AFBS spectral components passed through the optical filter ((3.1) or (3.2) in Figure 3); $\omega_i$ is the frequency of the "left" spectral component of the *i*-th AFBS; $\Omega_i$ is the address frequency; and $\varphi_{Ai}$ and $\varphi_{Bi}$ are the initial phases, which can be unequal, but their difference is constant over time.

The luminous power received by the photodetector from $N$ double-component AFBSs can be expressed by multiplying the Equation (1) with its complex conjugate:

$$P(t) = \left(\sum_{i=1}^{N} E_i(t)\right)\left(\overline{\sum_{i=1}^{N} E_i(t)}\right) = \left(\sum_{i=1}^{N}\left(A_i e^{j\omega_i t + \varphi_{Ai}} + B_i e^{j(\omega_i + \Omega_i)t + \varphi_{Bi}}\right)\right)\left(\sum_{k=1}^{N}\left(A_k e^{-(j\omega_k t + \varphi_{Ak})} + B_k e^{-(j(\omega_k + \Omega_k)t + \varphi_{Bk})}\right)\right) =$$

$$= \sum_{i=1}^{N}(A_i{}^2 + B_i{}^2) + 2\sum_{i=1}^{N} A_i B_i \cos(\Omega_i t + \varphi_{Ai} - \varphi_{Bi}) + 2\sum_{i=1}^{N}\sum_{k=i+1}^{N}\left(\begin{array}{l} A_i A_k \cos((\omega_i - \omega_k)t + \varphi_{Ai} - \varphi_{Ak}) + \\ A_i B_k \cos((\omega_i - \omega_k - \Omega_k)t + \varphi_{Ai} - \varphi_{Bk}) + \\ B_i A_k \cos((\omega_i - \omega_k + \Omega_i)t + \varphi_{Bi} - \varphi_{Ak}) + \\ B_i B_k \cos((\omega_i - \omega_k + \Omega_i - \Omega_k)t + \varphi_{Bi} - \varphi_{Bk}) \end{array}\right). \tag{2}$$

Thus, the oscillation of the amplitude of the electrical signal of the photodetector at the address frequency of the AFBS $\Omega_i$ is proportional to the amplitudes of the AFBS optical spectral components $A_i$ and $B_i$, which are defined by the parameters $u$ (the slope) and $v$ (the intercept) of the linear function describing the inclined frequency response of the optical filter ((3.1) or (3.2) in Figure 3):

$$A_i = L_0 \cdot (u \cdot \omega_i + v), B_i = L_0 \cdot (u \cdot (\omega_i + \Omega_i) + v), \tag{3}$$

where $L_0$ is the initial amplitude of the AFBS optical spectral components at the input of the filter with inclined linear frequency response. By measuring the amplitude of the photodetector output signal at the address frequency $\Omega_i$, it is possible to define the central frequency shift (or the frequency of the left spectral component $\omega_i$) of the AFBS relative to the inclined frequency response of the optical filter. However, due to the appearance of the additional frequency components in the last sum of Equation (2), the filtering of the electrical signal at the address frequencies is required.

By assuming that $B_i = A_i + L_0 \cdot u \cdot \Omega_i$ (which follows from (3)) and by filtering the photodetector output signal at the address frequency, the system of equations for the calculation of the AFBS spectral components' positions is obtained:

$$\sum_{i=1}^{N}\sum_{k=1}^{N}\left(\begin{array}{l} A_i A_k \cdot F(\Omega_j, \omega_i - \omega_k) + \\ A_i(A_k + u_k\Omega_k) \cdot F(\Omega_j, \omega_i - \omega_k - \Omega_k) + \\ A_k(A_i + u_i\Omega_i) \cdot F(\Omega_j, \omega_i - \omega_k + \Omega_i) + \\ (A_i + u_i\Omega_i)(A_k + u_k\Omega_k) \cdot F(\Omega_j, \omega_i - \omega_k + \Omega_i - \Omega_k) \end{array}\right) = D_j, \forall j = \overline{1, N}, \tag{4}$$

where the function $F(\Omega, \omega)$ describes the frequency response of the bandpass filter of the address frequency. The system of Equation (4) in the variable $\omega_i$ is solved numerically (for example, using the Levenberg–Marquardt or the Newton–Raphson algorithms), taking the previously calculated value of $\omega_i$ as the initial conditions.

In most cases, the error of the AFBS spectral position calculation does not exceed 0.1 pm even in multi-sensor systems comprising double-component structures [18]. However, in certain cases of the AFBS spectral components' relative positions, when the components of different AFBSs coincide or the third summand in (2) coincides with the address frequency of any of the structures, the error can reach 2 pm [18]. This issue can be mitigated with the usage of the addressed structures having three or more spectral components forming two or more address frequencies [14].

As is known, fiber Bragg gratings are sensitive both to strain and temperature at the same time; therefore, FBG-based sensor systems generally include at least one FBG isolated from any physical influence except for the temperature in order to perform thermal compensation of the other sensors. Thus, it is necessary to define a method for the combined calibration of strain and temperature sensors. In the work [19], the following procedure is described, considering a combination of strain and temperature sensors as an example. The value of strain $\varepsilon$ causing the central wavelength shift $\Delta\lambda_\varepsilon$ of an AFBS can be expressed as a third power polynomial [20]:

$$\varepsilon = a_3 \cdot \Delta\lambda_\varepsilon^3 + a_2 \cdot \Delta\lambda_\varepsilon^2 + a_1 \cdot \Delta\lambda_\varepsilon + a_0, \tag{5}$$

where $a_i$, $i = 1 \ldots 3$ are certain coefficients. In its turn, the value of temperature $T$ causing the wavelength shift $\Delta\lambda_T$ of the sensor is defined as a second power polynomial [20]:

$$T = c_2 \cdot \Delta\lambda_T^2 + c_1 \cdot \Delta\lambda_T + c_0, \tag{6}$$

where $c_i$, $i = 1, 2$ belong to the other set of coefficients. In order to take into account the thermal wavelength shift of the strain sensor, the coefficients $a_0 \ldots a_3$ in (5) can be expressed as the functions of temperature, similarly to (6):

$$\varepsilon = a_3(T) \cdot \Delta\lambda_\varepsilon^3 + a_2(T) \cdot \Delta\lambda_\varepsilon^2 + a_1(T) \cdot \Delta\lambda_\varepsilon + a_0(T), \tag{7}$$

where

$$a_n(T) = c_{2,n} \cdot \Delta\lambda_T^2 + c_{1,n} \cdot \Delta\lambda_T + c_{0,n}, n = \{0, 1, 2, 3\}. \tag{8}$$

The resulting strain dependence on the central wavelength shifts ($\Delta\lambda_T$, $\Delta\lambda_\varepsilon$) can be expressed as follows:

$$\varepsilon = F(\Delta\lambda_T, \Delta\lambda_\varepsilon, c_{m,n}) = \sum_{m=0}^{2} \sum_{n=0}^{3} c_{m,n} \cdot \Delta\lambda_T^m \cdot \Delta\lambda_\varepsilon^n . \tag{9}$$

During the sensor calibration, the dataset $\{\Delta\lambda_{Ti}, \Delta\lambda_{\varepsilon i}, T_i, \varepsilon_i\}$ is obtained, in which $\Delta\lambda_{Ti}$ is the central wavelength shift of the temperature sensor, $\Delta\lambda_{\varepsilon i}$ is the central wavelength shift of the strain sensor, $T_i$ and $\varepsilon_i$ are the values of temperature and strain induced by the thermal chamber and the translation stage, respectively, and $i$ is the number of test measurements. The unknown coefficients $\{c_{m,n}\}$ of the approximating surface (9) are defined from the conditions of the minimal deviation of the measured dataset $\{\Delta\lambda_{Ti}, \Delta\lambda_{\varepsilon i}, T_i, \varepsilon_i\}$ from this approximating surface. The unknown coefficients $\{c_{m,n}\}$ are calculated using the least squares method so that the surface (9) follows the strain sensor behavior as precisely as possible at various strain–temperature combinations [19], i.e., the condition is met:

$$\Phi = \sum_{i=1}^{N} (\varepsilon - \varepsilon_i)^2 = \sum_{i=1}^{N} \left( F(\Delta\lambda_T, \Delta\lambda_\varepsilon, c_{m,n}) - \varepsilon_i \right)^2 \to \min. \tag{10}$$

The minimum condition of (10) requires all partial derivatives of the function $\Phi$ with respect to all of the $\{c_{m,n}\}$ to be equal to zero:

$$\frac{\partial\Phi(c_{m,n})}{\partial c_{m,n}} = 2 \sum_{i=1}^{N} \left[ \left( F(\Delta\lambda_T, \Delta\lambda_\varepsilon, c_{m,n}) - \varepsilon_i \right) \cdot \frac{\partial F(\Delta\lambda_T, \Delta\lambda_\varepsilon, c_{m,n})}{\partial c_{m,n}} \right] = 0 \; \forall m \in \{2, 1, 0\} \cup n \in \{3, 2, 1, 0\}. \tag{11}$$

Thus, the system of 12 equations is obtained, by solving which the 12 unknown coefficients $\{c_{m,n}\}$ are calculated.

*Challenges and limitations.* The main limitation to the number of AFBS sensors that can be simultaneously interrogated in the system is the maximum operating frequency of the photodetector, i.e., the maximum address frequency, at which the beating signal can be generated by the photodetector. One of the main components of the measurement error is the deviation of the optical filter frequency response from its linear approximation, since the amplitudes of the AFBS spectral components used for the calculation of the AFBS spectral position relative to the optical filter are defined by the parameters of the linear function describing the inclined frequency response of the filter. Another challenge is the necessity to ensure the uniformity of strain and temperature impact on the AFBS sensing element in order to maintain its address frequency unchanged.

The references regarding the subject of AFBS interrogation are listed in Table 1.

**Table 1.** Works related to the AFBS interrogation.

| References | Subject |
|---|---|
| [17] | Requirements to the optical filter with linear frequency response; compensation of the thermal drift of the optical filter. |
| [18] | Accuracy estimation of the AFBS Bragg wavelength determination. |
| [14] | Interrogation of multi-addressed fiber Bragg structures. |
| [19,20] | Combined calibration of strain and temperature sensors. |

## 4. Fabrication Methods of Addressed Fiber Bragg Structures

This section considers the fabrication techniques for transmitting ($N\pi$-FBG) and reflecting ($N\lambda$-FBG) AFBS separately. As an example, the fabrication of asymmetric triple-component AFBSs is discussed.

### 4.1. Fabrication of Transmitting AFBS

For the fabrication of the transmitting addressed structures with ultra-narrowband transparency windows in their spectra, the following techniques were chosen: the technique involving the spectral overlapping of two identical FBGs [21], the technology based on a stepped phase mask [22] and the method based on changing the geometry of an optical fiber using an electric arc of a welding machine [23]. The works [21,23] use conventional phase masks to record FBGs, while a special mask with a thickness difference of 2300 nm was utilized in [22], using which a phase shift was formed at the place of the mask thickness step.

Based on the abovementioned methods, the following technique for $N\pi$-FBG fabrication has been established [24]. In a single-mode optical fiber (for example, SMF-28), an FBG is recorded using conventional holographic recording schemes based on the Lloyd's interferometer [21]. After that, the fiber is shifted transversely to the recording beam using a high-precision translation stage, and another FBG is recorded. The displacement and the beam size are chosen in such a way that the FBGs are superimposed; thus, a phase shift is created and a $2\pi$-FBG-type structure with length $L_1$ is formed. For more accurate control of the shift value, the scheme can use a Michelson interferometer based on bulk optical elements [21]. Then, in accordance with the technique presented in [23], the fiber is displaced by a distance equal to the first $2\pi$-FBG length ($L_1$) and the distance between the gratings ($\Delta l$), after which the second $2\pi$-FBG with the length $L_2 \neq L_1$ is recorded using the technology [21]. Since the recording conditions remain unchanged, the characteristics of both $2\pi$-FBGs will be basically the same, except for a slight difference in the bandwidth of their spectra. At the next step, the fiber is placed in a splicer, and the electric arc induces a phase difference between the radiation reflected from both structures, thus creating a $3\pi$-FBG structure, as shown in Figure 5. The bandwidth of the AFBS transparency windows created using the presented method is 30–35 MHz [23].

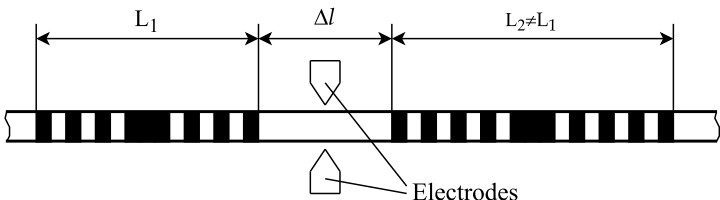

**Figure 5.** The structure of an asymmetric $3\pi$-FBG fabricated using a combination of techniques in [21,23].

The work [25] studies the influence of the $N\pi$-FBG structure parameters on the spectral response of such structures. Thus, an increase in the induced refractive index and the length of the outermost uniform sections of the AFBS has the greatest effect on reducing the width of the transparency windows. Asymmetric change of address frequencies are achieved by varying the phase shifts. The findings of [25] can be used to obtain the required spectral characteristics of the $N\pi$-FBG structure.

### 4.2. Fabrication of Reflecting AFBS

The reflecting $N\lambda$-FBG addressed structures can be fabricated by sequential recording of $N$ conventional FBGs with different Bragg wavelengths using a standard setup based on a 244 nm ultraviolet laser and the nanometer translation stages, such as STANDA-8MT173 [26,27].

The work [28] presents a method suitable for $N\lambda$-FBG fabrication using strain of an optical fiber and a phase mask. A translation stage is used to move the phase mask, while a spring induces tension on the optical fiber. Using the presented method, two identical FBGs were recorded with the difference in the Bragg wavelength of 1 nm and a bandwidth of 0.3 nm [28]. It is possible to obtain narrower grating bandwidths by increasing their lengths, as we demonstrate in Figure 6, where the spectral response of a 2$\lambda$-FBG with the component bandwidths of 110 pm is shown.

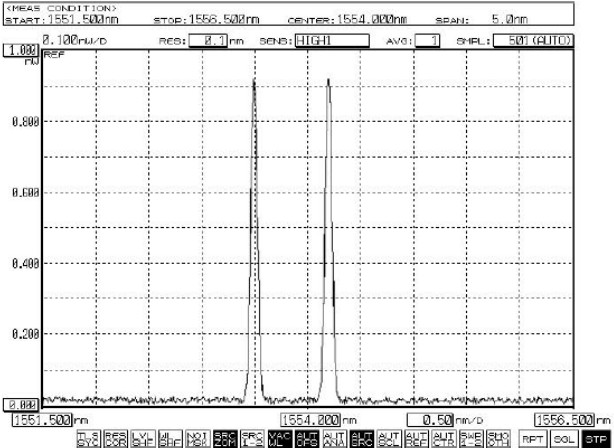

**Figure 6.** Reflectance spectrum of a 2$\lambda$-FBG recorded at KNRTU-KAI.

Another method for $N\lambda$-FBG fabrication uses structured grating technology by alternating sections with an induced grating and "empty" sections of the fiber [29]. During the recording, the phase mask is fixed, and the fiber is displaced with an accuracy of 10 nm. At certain fiber positions, the recording ultraviolet beam is turned off. The parameters of the resulting $N\lambda$-FBG structure are adjusted by varying the displacement step of the translation stage. In [29], a cascaded connection of structured FBGs with estimated bandwidths of 1 pm was proposed.

Thus, it is recommended to use the $N\lambda$-FBG structures recorded with the conventional phase mask method in the applications with a small number of sensors, while the technology of structured FBG recording should be used for the sensors applied in high-resolution measurement systems with a large number of sensors due to the narrower bandwidths of the spectral components.

*Challenges and limitations.* The lowest address frequency of an AFBS is determined mainly by the width of its spectral components; therefore, it is preferable to ensure the smallest bandwidth of such components of the fabricated AFBS. The latter is achieved by increasing the induced refractive index in the case of transmitting AFBS, and by decreasing the induced refractive index and increasing the FBG length at the same time in the case of reflecting AFBS. The former is limited by the photosensitivity of the optical fiber, while the latter is restricted by the maximum length of the sensing element suitable for the particular application.

The references regarding the subject of AFBS fabrication are listed in Table 2.

**Table 2.** Works related to AFBS fabrication.

| References | Subject |
| --- | --- |
| [21] | Recording of phase-shifted FBGs by physical overlapping. |
| [22] | Recording of phase-shifted FBGs using stepped phase mask. |
| [23] | Recording of phase-shifted FBGs by changing the geometry of an optical fiber using an electric arc. |
| [24] | $N\pi$-FBG fabrication using a combination of FBG physical overlapping and changing the fiber geometry techniques. |
| [25] | Influence of the $N\pi$-FBG structure parameters on its spectral response. |
| [26,27] | Sequential recording of FBGs. |
| [28] | Reflecting AFBS recording using strain of an optical fiber and a phase mask. |
| [29] | Reflecting AFBS recording using structured technology by alternating sections with an induced FBG and "empty" fiber sections. |

## 5. Application of Addressed Fiber Bragg Structures

In this section, several examples of AFBS application are discussed. Generally, the AFBSs can be used in various systems instead of conventional fiber Bragg grating sensors, providing such advantages as a simplified interrogation scheme and enhanced metrological performance, including higher measurement resolution and rate. It must be noted that the $N\lambda$-FBG structures are commonly longer than the $N\pi$-FBGs, which limits the applicability of the former in the areas where the sensor length is restricted. On the other hand, the $N\lambda$-FBGs can be connected sequentially and recorded in the same optical fiber, thereby providing more flexibility in the system layout in comparison with the $N\pi$-FBGs that require parallel connection.

### 5.1. Power Engineering

The features of fiber-optic sensors, namely immunity to electromagnetic interference, ability to operate in harsh environments, suitability for remote sensing, intrinsic galvanic isolation, etc., make them advantageous in various monitoring systems for power engineering [30]. An example of such application is the vibration monitoring of underground power transmission lines, which is necessary for detecting destructive vibrations and preventing the damages. The systems for simultaneous vibration and temperature measurement are of particular interest for power transmission lines monitoring. Such distributed acoustic sensor (DAS) systems can be based on a distributed array of a large number of FBGs. However, the common disadvantages of the DAS systems based on an array of ultra-weak FBGs [31,32] are the high cost of optoelectronic interrogators and the complexity of the interference device, its calibration and control. In order to eliminate these issues, the work [33] introduced the microwave-photonic DAS system based on $N\lambda$-FBG addressed structures.

Another proposed application of AFBS in power engineering is the measurement of the relative humidity of switchgear devices [34]. Relative humidity (RH) is an important parameter that is used to determine the possibility of the intensity increase in partial discharges, formation of an arc, etc. RH sensors can be developed using an FBG with hygroscopic coating instead of the standard silicone covering. The most widely used coating material for RH sensors is polyimide [35,36], which provides the sensitivity of 2–6 pm/%. Figure 7 presents an RH sensor developed in [34] consisting of three $2\pi$-FBG AFBSs: AFBS$_1$ is a strain sensor with polyimide coating (binding areas are shown in checkered filling, and flexible zones are highlighted with oblique filling), AFBS$_2$ with etched cladding acts as a refractometer sensor, while AFBS$_3$ acts as a temperature sensor.

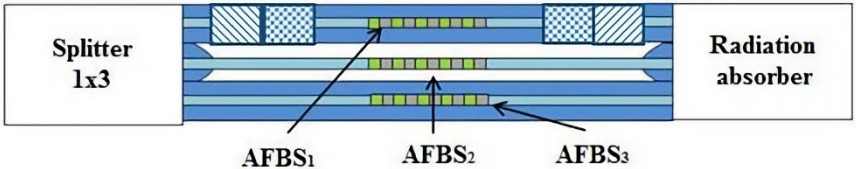

**Figure 7.** Scheme of an RH sensor based on three AFBSs [34].

As shown in [34], the Bragg wavelength of the AFBS$_1$ with polyimide coating linearly increases with the increase in the RH at constant temperature, due to the strain effect caused by the expansion of the polyimide when it absorbs the moisture.

In addition to RH measurement, the proposed sensor provides measurements of condensed moisture, which are also important in high-voltage power applications, by means of a refractometric sensor with an etched cladding (AFBS$_2$). The AFBS$_2$ is sensitive to the refractive index of the environment due to the etched cladding. The increase in the environmental refractive index causes the increase in the Bragg wavelength of the AFBS$_2$ [37], thereby enabling the sensor to detect the condensed moisture. The sensor's temperature measurement range was $-60 \ldots + 180\,^{\circ}\text{C}$ with an error of $\pm 0.1\,^{\circ}\text{C}$, and it demonstrated the sensitivity of RH measurement of 6 pm/% in the range of $30\% \ldots 80\%$ [34]. By multiplexing a number of such RH sensors, it is possible to create a multi-sensor system for distributed monitoring of RH, the amount of condensed moisture, the level of partial discharges, the temperature of buses, contacts and other elements of power systems.

The references regarding the subject of AFBS application in power engineering are listed in Table 3.

**Table 3.** Works related to the AFBS application in power engineering.

| References | Subject |
| --- | --- |
| [30] | Overview of FBG applications in power industry. |
| [31,32] | Distributed acoustic sensor system based on an array of ultra-weak FBGs. |
| [33] | Distributed acoustic sensor system based on AFBS. |
| [34] | Measurement of relative humidity of switchgear devices using AFBS. |
| [35,36] | Measurement of relative humidity using FBG with polyimide coating. |
| [37] | Detection of environmental refraction index change using FBG with etched cladding. |

### 5.2. Oil and Gas Industry

Fiber-optic sensors are widely used in the oil and gas industries due to their inherent advantages, namely the ability to operate in harsh environments, including at temperatures exceeding 250 °C, the possibility to create systems for distributed and remote measurements over long distances, as well as the lack of electrical power supply in sensing locations [5]. Distributed fiber-optic sensor systems based on Raman and Brillouin scattering [38,39] have been used for thermal monitoring, by means of which, for example, pipeline leak detection can be performed. In addition, pipeline structures are prone to damage due to ground movement caused by earthquakes, erosion, etc., and the strain measurement technique based on Brillouin scattering has been used to monitor the movement of the pipeline [39].

The work [40] proposes a combined sensor system for simultaneous local and distributed strain and temperature measurements [41] for downhole telemetry. The scheme of the system is presented in Figure 8 [40].

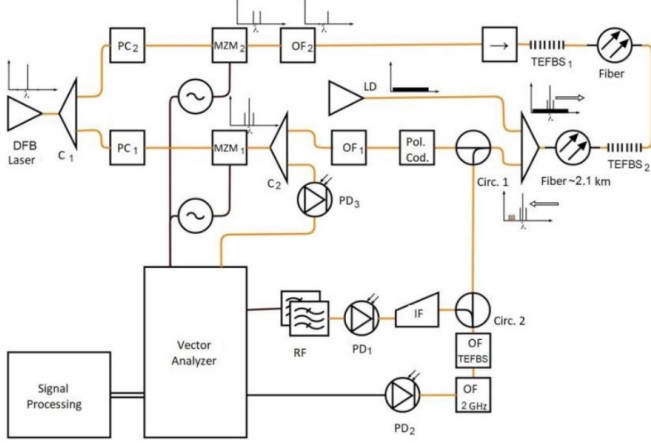

**Figure 8.** Scheme of combined sensor system for local and distributed downhole monitoring [40].

The system includes a DFB laser generating continuous narrowband radiation, which is divided into two branches by means of fiber-optic coupler $C_1$. In one of the branches, a continuous signal for Brillouin pumping is formed, while in the other branch, a continuous temperature probe is created. A wideband optical radiation from another source (LD) is directed to the addressed structures $TEFBS_1$ and $TEFBS_2$ (referred to as double-component AFBS (DCAFBS) in the current paper), which serve as local temperature sensors. The radiation from LD then enters the fiber from the side of the pump arm after the circulator Circ. 1. The polarization controller $PC_1$ and the Mach–Zehnder modulator $MZM_1$ controlled by a vector network analyzer (vector analyzer) are utilized for sinusoidal modulation of the continuous radiation intensity from the DFB laser. Noise is removed by an optical filter $OF_1$, and a polarization coder (pol. cod.) depolarizes the pump signal and allows the avoidance of fluctuations caused by polarization in Brillouin amplification. In the second branch, the optical radiation is modulated by another modulator $MZM_2$ in accordance with a microwave signal generator to create a probing signal. After that, optical filter $OF_2$ selects the low-frequency probe (Stokes component), removes noise as well as the secondary carrier, and also selects a high-frequency probe sideband (anti-Stokes component). The pump laser radiation passes through an optical circulator (Circ. 1) and enters a ~2.1-km standard single-mode fiber (SMF-28), which is also used to select both the probe signal and the reflected components at the pump frequency. Both radiation components are directed to the optical circulator (Circ. 2), after which the addressed structures' components are selected with the filter OF TEFBS and are transmitted through the filter with inclined frequency response (IF) and received by the photodetector $PD_1$, while the reflected pump light filtered by 2 GHz OF is received by $PD_2$. The work [40] reported that the system allows one to simultaneously and independently carry out distributed and local temperature measurements with the resolution of 0.01 °C, providing the benefits of addressed interrogation of local sensors.

The references regarding the subject of AFBS application in the oil and gas industries are listed in Table 4.

**Table 4.** Works related to the AFBS application in oil and gas industries.

| References | Subject |
|---|---|
| [5] | Overview of applications of fiber-optic sensors in oil and gas industries. |
| [38,39] | Thermal and strain monitoring of pipelines using distributed fiber-optic sensors. |
| [40] | AFBS-based system for combined local and distributed sensing for downhole telemetry. |
| [41] | Combined sensor system based on Brillouin optical frequency-domain analysis and FBG for simultaneous temperature/strain measurements. |

*5.3. Automotive Engineering*

The significantly reduced costs of AFBS interrogation systems due to their significant simplification (as it was described in Section 3), in comparison with the traditional optoelectronic FBG interrogators, open up opportunities for FBG-based sensor application in areas where it has not been economically viable, such as in automotive engineering.

One of the examples of such AFBS applications is the instrumentation of the load-sensing wheel hub bearings [13]. The load-sensing bearing is a promising type of the automotive sensing components that enhance the efficiency of various active safety systems of vehicles [42]. For instance, when the vehicle brakes or accelerates, the load sensing wheel hub bearings detect the maximum wheel load, and the wheel slip ratio is corrected in such a way as to maintain the peak value of the wheel load, therefore enhancing the effectiveness of the wheel–road interaction [43].

The load-sensing wheel hub bearing operates by measuring the strain of the bearing outer ring, which is used to estimate the load applied to the bearing. The research conducted in [13] demonstrated the feasibility of AFBS usage as sensing elements in automotive load-sensing bearings. Figure 9 shows an experimental setup for static load tests including a prototype bearing with two AFBS sensors. The first AFBS acts as a strain sensor detecting the tangential deformation of the bearing outer ring, and the second AFBS (not shown in Figure 9) is isolated from strain and serves for thermal compensation of the strain sensor.

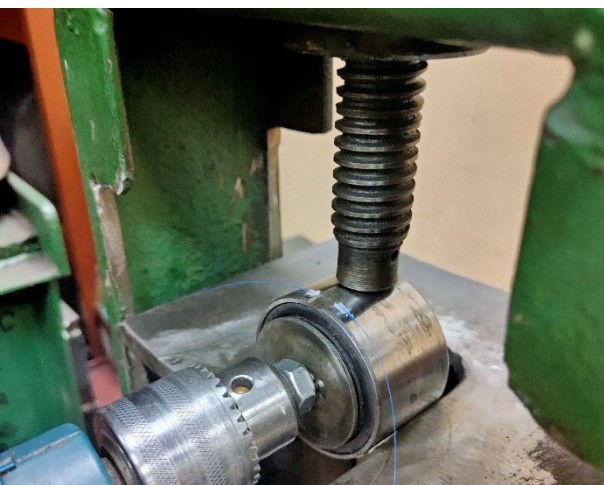

**Figure 9.** Automotive load-sensing bearing with AFBS sensor during static load testing [13].

In this application, minimal sensor length is required, therefore a $2\pi$-FBG structure was used with a typical length of 5–7 mm, since the $N\pi$-FBGs are generally shorter in comparison with $N\lambda$-FBGs. This is due to the fact that in order to ensure the narrow bandwidth of the FBG spectrum, its induced refractive index must be low, which in its turn necessitates the increase in the FBG length to keep the resulting FBG reflectance sufficiently high.

Another development direction of the automotive sensor systems are the so-called "intelligent tires", which provide real-time measurements of tire grip parameters, including tire–road friction, contact patch dimensions, loads, etc., in various driving conditions [44]. Fiber-optic sensors also possess benefits in intelligent tire instrumentation due to their small footprint, low weight, flexibility and the possibility to be embedded in the tire structure. Figure 10 shows an FBG-based strain sensor attached to the inner surface of a prototype intelligent tire [44].

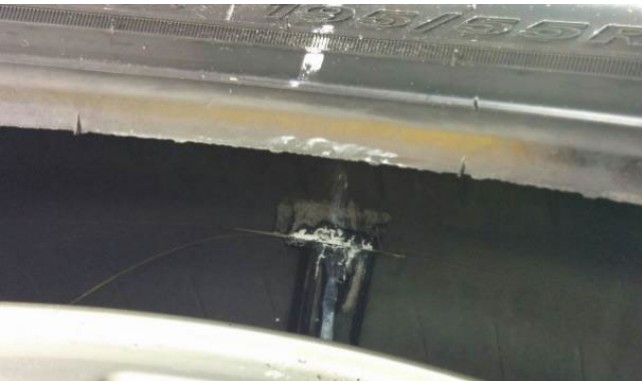

**Figure 10.** FBG-based strain sensor attached to the inner surface of the tire [44].

The works [19,45] proposed the usage of AFBSs instead of conventional FBGs to measure the strain and temperature of intelligent tires. An additional advantage of the simplified interrogation scheme of AFBS, besides the reduced costs, is the possibility to create a compact and vibration-proof interrogator suitable for in-wheel installation. In the proposed system, a flexible ring model [46] is used to analyze the tangential deformation of the tire.

The works regarding the subject of AFBS application in automotive engineering are listed in Table 5.

**Table 5.** Works related to the AFBS application in automotive engineering.

| Reference | Subject |
|---|---|
| [13] | Instrumentation of load-sensing wheel hub bearing using AFBS. |
| [42] | Application of load-sensing wheel hub bearings for vehicle dynamics control. |
| [44] | Instrumentation of "intelligent" tires using FBGs. |
| [19,45] | Instrumentation of "intelligent" tires using AFBSs. |

### 5.4. Medicine

Fiber-optic sensors also offer benefits in a range of biomedical applications due to their small cross-section, lightness, flexibility and chemical resistance, which makes them minimally invasive and suitable for in vivo measurements (i.e., measurements directly inside a patient) [4]. Pressure measurement is an important asset in various medical procedures, including cardiovascular, urologic diagnostics and the monitoring of invasive treatments. A significant research effort is dedicated to the development of catheters for high-resolution manometry, which presupposes the spatial resolution of pressure measurement of 1–2 cm [47].

Addressed fiber Bragg structures were utilized in a catheter for esophagus sphincters monitoring proposed in [48]. The scheme of the catheter placement is shown in Figure 11.

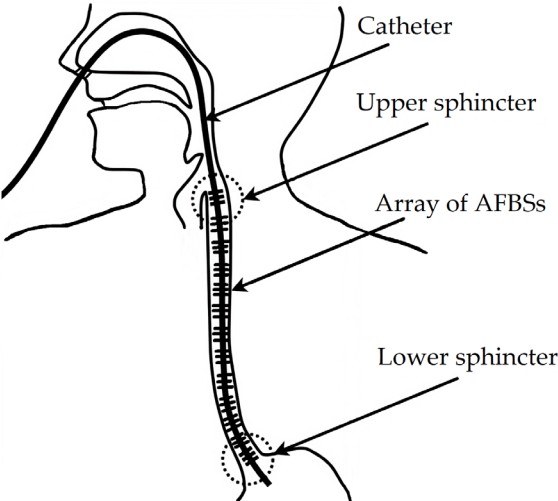

**Figure 11.** Scheme of catheter placement for esophagus sphincters monitoring, according to [48].

The catheter is instrumented with an array of $2\pi$-FBG addressed structures that measure pressure caused by the sphincters of the esophagus. The interrogation of AFBSs is implemented according to the scheme described in Section 3 of the current paper, which provides benefits in terms of implementation costs in comparison with the conventional FBG sensors. The experimental investigation of the catheter was carried out taking the values of pressure measured by certified insufflator for reference. The maximum deviation of the pressure values measured with the catheter was $\pm$0.1%, which meets the requirements for medical applications [48]. In addition, a catheter with a larger number of AFBS sensors (up to 72) was proposed for intestinal peristalsis monitoring [48].

The references related to the subject of AFBS applications in medicine are listed in Table 6.

**Table 6.** Works related to the AFBS application in medicine.

| Reference | Subject |
|---|---|
| [4] | Application of fiber-optic pressure sensors in medicine. |
| [47] | Instrumentation of high-resolution catheters using FBGs. |
| [48] | Instrumentation of high-resolution catheters using AFBSs. |

*5.5. Environmental Monitoring*

A significant research effort is now dedicated to the development of various sensors for environmental monitoring, particularly for concentration measurements of greenhouse gases. An attractive solution for simultaneous gas, temperature and pressure measurement is based on the combination of FBG and a Fabry–Perot resonator (FPR). Such combined fiber-optic sensors (CFOSs) consist of an FPR formed of a thin film at the end face of the optical fiber with FBG near it [49]. The film is made of a polymer, the refractive index of which reversibly changes depending on the gas concentration. The spectral response of the FPR is sensitive to various environmental parameters, including the changes of gas concentration, temperature and pressure, while the FBG central wavelength shift depends mainly on the temperature, which enables the simultaneous measurement of temperature and gas concentration using the CFOS.

The usage of AFBS instead of conventional FBG in CFOS has the advantage of a simplified interrogation scheme as well as the possibility to include several AFBSs with identical central wavelength and addressable interrogation. The scheme of the addressed combined fiber-optic sensor (ACFOS) incorporating a 2λ-FBG structure is presented in Figure 12 [49]. The thickness of the film $h$ and its material are selected based on the type of gas under test. For instance, PEI/PVA coating is used for $CO_2$ concentration measurements [50]; polyaniline/$Co_3O_4$–for CO [51]; $LuPc_2$–for $NO_2$ [52]; PDMS/PMMA– for $NH_3$ [53]; Cryptophane A–for $CH_4$ [54] and others. Humidity measurement can be performed using PVA coating [55].

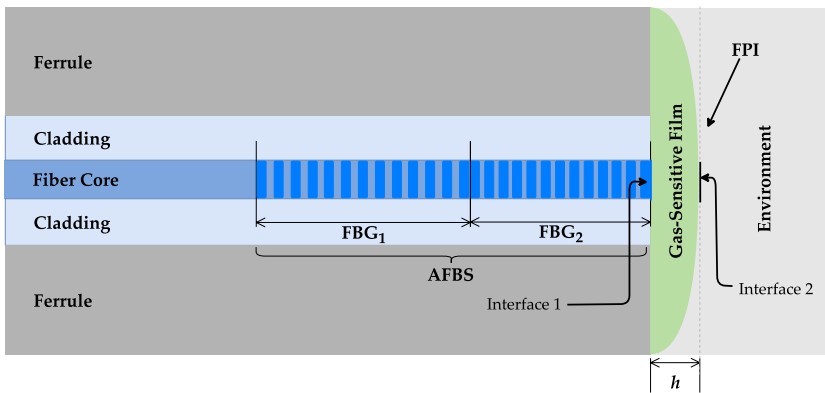

**Figure 12.** The structure of ACFOS [49].

The ACFOS spectrum is a superposition of the AFBS and FPR spectra, as shown in Figure 13; therefore, it is necessary to separate the FBG and the FPR spectra in order to successfully interrogate the combined sensor, which can be achieved using Karhunen– Loeve transform [56]. According to estimations, the sensor system based on ACFOS allows the measuring the gas concentrations in the range of 10−90% with an error of 0.1−0.5% [49].

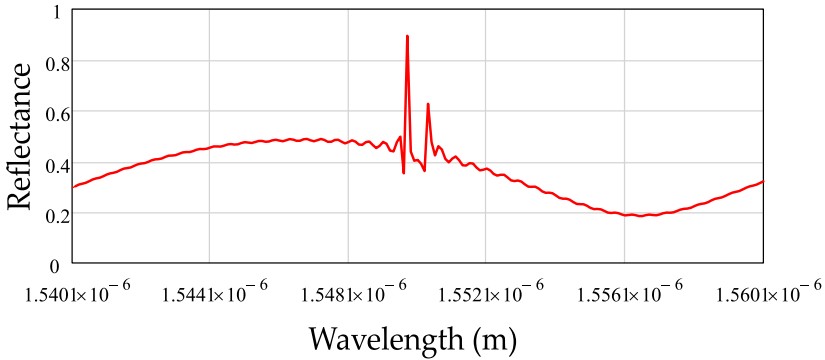

**Figure 13.** ACFOS reflectance spectrum [49].

The works regarding the subject of AFBS application in environmental monitoring are listed in Table 7.

**Table 7.** Works related to AFBS application in environmental monitoring.

| References | Subject |
|---|---|
| [49] | AFBS usage in combined sensors for gas concentration measurement. |
| [50] | Fiber-optic Fabry–Perot interferometer based on PEI/PVA coating for $CO_2$ concentration measurement. |
| [51] | Fiber-optic Fabry–Perot interferometer based on polyaniline/$Co_3O_4$ coating for CO concentration measurement. |
| [52] | Fiber-optic Fabry–Perot interferometer based on $LuPc_2$ coating for $NO_2$ concentration measurement. |
| [53] | Fiber-optic Fabry–Perot interferometer based on PDMS/PMMA coating for $NH_3$ concentration measurement. |
| [54] | Fiber-optic Fabry–Perot interferometer based on Cryptophane A coating for $CH_4$ concentration measurement. |
| [55] | Fiber-optic Fabry–Perot interferometer for humidity measurement. |
| [56] | Interrogation of combined sensors using Karhunen–Loeve transform. |

## 6. Development Prospects of Addressed Fiber Bragg Structures

Based on the theoretical and implementation background of the addressed fiber Bragg structures reviewed above, several shortcomings of the current AFBS techniques can be identified. The disadvantage of the $N\lambda$-FBG addressed structures, in general, is that they have significantly greater length in comparison with the $N\pi$-FBGs, which makes it problematic to ensure the uniform temperature and strain variations of all the FBGs constituting the $N\lambda$-FBG in order to maintain the unchanged address frequencies. In addition, some applications require the usage of AFBSs with high difference frequencies between their spectral components (up to hundreds of GHz). In order to obtain the corresponding beating frequency of the electrical signal, a high-speed photodetector is required, which significantly increases the total cost of the system. However, a number of directions of AFBS further development can be drawn that mitigate the abovementioned shortcomings of the current AFBSs' state of the art and expand the possibilities of their application.

### 6.1. Moiré Recording of Nλ-FBG

As stated before, the $N\lambda$-FBG addressed structures are formed by sequential recording of several ultra-narrowband FBGs with different central wavelengths. In order to provide ultra-narrowband spectral response, it is necessary to increase the FBG length, and the sequential recording of a number of such FBGs results in the total length of $N\lambda$-FBG reaching centimeters or even dozens of centimeters. However, in order to maintain the unchanged address frequency, it is crucial to ensure that all the FBGs belonging to the same AFBS are subjected to the same strain or temperature impact, which is problematic in cases of extensive structure length. This issue can be mitigated by using the moiré recording technique [57,58]. During the moiré fabrication, the first FBG is written using the conventional recording method, such as the one using phase mask, then the fiber is stretched, and the second FBG is written directly over the first grating. Stretching the fiber between the recordings changes the length (and the period) of the initial grating with respect to the second. Since both FBGs are located in the same section of the optical fiber, they are always subjected to the same temperature and strain, therefore the address frequency of such $2\lambda$-FBG remains invariant.

### 6.2. Transverse Load Sensing Using 2π-FBG

Fiber Bragg gratings can be used not only to measure axial strain of the optical fiber but also to measure the transverse load applied to it, which is possible due to the induced-birefringence effects of the FBGs [59]. Axial strain results in linear shift of the FBG central wavelength, while the transverse load causes an additional birefringence in the optical fiber, creating two different Bragg wavelengths corresponding to each of the polarization modes. This effect can be utilized to measure transverse load since the

wavelength difference between the two Bragg wavelengths has a linear relationship with the applied load. However, due to the significantly lower sensitivity to the transverse load in comparison with the axial load sensitivity, this limits the measurement resolution of the transverse load of the conventional FBGs, which have much wider spectral bandwidth than the wavelength shift induced by the birefringence. For this reason, the usage of the FBG with phase shift provides more accurate measurements of the transverse load because of the narrowness of the transmission window of such FBG [60].

If a $2\pi$-FBG structure is subjected to the transverse load, then the output optical radiation includes four components:

$$I(t) = I_{-1-1}\cos(\omega_{-1-1}t) + I_{-1+1}\cos(\omega_{-1+1}t) + I_{+1-1}\cos(\omega_{+1-1}t) + I_{+1+1}\cos(\omega_{+1+1}t), \quad (12)$$

where $\omega_{-1-1}$, $\omega_{-1+1}$, $\omega_{+1-1}$, $\omega_{+1+1}$ and $I_{-1-1}$, $I_{-1+1}$, $I_{+1-1}$ and $I_{+1+1}$ are the frequencies and the amplitudes of the spectral components of the optical signal formed due to the induced-birefringence effect. The address frequency $\Omega$ is equal to the difference between the maximum beating frequency and the half of the minimal beating frequency:

$$\Omega = (\omega_{+1+1} - \omega_{-1-1}) - (\omega_{+1+1} - \omega_{+1-1})/2. \quad (13)$$

The frequency resulting from the transverse load corresponds to the minimal beating frequency. Therefore, the main requirement of the $2\pi$-FBG for the transverse load sensing is the minimal beating frequency at the maximum transverse load being lower than half of the address frequency, i.e., $(\omega_{+1+1} - \omega_{+1-1}) < \Omega$.

### 6.3. Address Frequency Downshifting

In some applications, it is necessary to use the AFBS with high address frequency to ensure the sufficient difference of the spectral components' amplitudes. An example of such application is the probing of the spectral response of the Fabry–Perot interferometer (FPI) in order to define its position relative to the AFBS spectrum, which can also be used in gas concentration measurements. In Figure 14, the dashed line represents a spectral response of the Fabry–Perot interferometer, while the solid lines denote the spectral components of the AFBS. In order to provide the sufficient difference of the AFBS spectral components' amplitudes, the difference frequency $\Omega_3$ must be rather high, reaching hundreds of GHz, due to the slow variation of the FPI amplitude over frequency. Therefore, the usage of the double-component AFBS in this application is problematic since it requires the usage of a high-frequency photodetector to generate the beating frequency of hundreds of GHz. This issue can be solved by using the AFBS with four spectral components, as shown in Figure 14. The difference frequency between the first and the second component ($\Omega_1$), as well as the third and the fourth one ($\Omega_2$) are in the range of several GHz, which is significantly lower than $\Omega_3$. Using the amplitudes of the beating frequencies $\Omega_1$ and $\Omega_2$ and knowing the difference frequency $\Omega_3$, it is possible to define the position of the FPI spectrum relative to the AFBS spectral response, using a comparatively low-frequency photodetector.

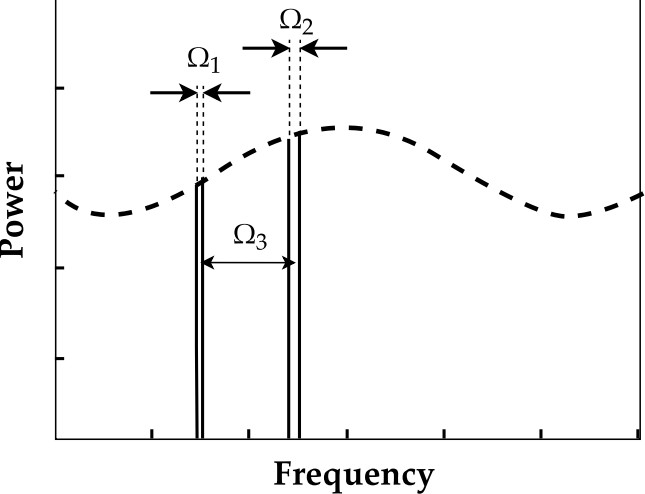

**Figure 14.** Probing of the Fabry–Perot interferometer using AFBS.

The proposed "downshifting" of the address frequency can also be implemented in the cases when it is required to separate the spectral components by high frequency in order to ensure that they do not coincide with the components of another AFBS.

## 7. Conclusions

The current paper is dedicated to comprehensive review of the works on the subject of addressed fiber Bragg structures (AFBS) covering theoretical background, interrogation principles, fabrication, calibration, as well as implementation examples. Generally, AFBS can be used in all the applications of conventional FBG-based sensors offering a simplified interrogation scheme, which can provide additional benefits in some areas, such as increased measurement rate and resolution. Based on the presented review, the directions of AFBS further development are proposed, including the moiré recording of the $N\lambda$-FBG structures and transverse load sensing using AFBSs with phase shifts.

**Author Contributions:** T.A. and A.S. drafted the manuscript. O.M. supervised the work. T.A., G.I., A.K., R.M. (Rinat Misbakhov), R.M. (Rustam Misbakhov), G.M., O.M., I.N. and A.S. commented, edited and reviewed the manuscript. All authors have read and agreed to the published version of the manuscript.

**Funding:** This work was financially supported by the Ministry of Science and Higher Education as part of the "Priority 2030" program.

**Institutional Review Board Statement:** Not applicable.

**Informed Consent Statement:** Not applicable.

**Data Availability Statement:** The data presented in this study are available on request from the corresponding author. The data are not publicly available due to rules of our contract conditions with our customer.

**Conflicts of Interest:** The authors declare no conflict of interest.

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
