# Peer review of "Overview of Addressed Fiber Bragg Structures’ Development"

_photonics, doi:10.3390/photonics10020175_

Round 1

Reviewer 1 Report

Please, see attached file.

Author Response

First of all, we would like to thank the Reviewer for the precious comments that allowed us to improve the quality of the paper.

The authors present an overview of fabrication, theoretical description and applications of addressed fiber Bragg structures. The authors have done a good job. However, before manuscript publication, I recommend the authors to consider the following points:

  1. Line 181:

It seems that L0 is missing in the equation Bi = Ai + u · Ωi. It may be the right equation is Bi = Ai + L0 · u · Ωi. The authors would consider checking it.

Thank you for the concern. The equation was corrected.

  1. Line 396:

I believe that the word "baring" was written instead of "bearing" (caption of figure 8).

Thank you for the remark. The misprint was corrected.

  1. Lines from 522 to 525:

“An example of such application is the probing of the spectral response of the Fabry-Perot interferometer (FPI) in order to define its position relative to the AFBS spectrum, which can also be used in gas concentration measurements". Maybe a reference is missing here. Perhaps, the authors would consider citing a reference.

Thank you for the suggestion. This section considers development prospects of addressed fiber Bragg structures, and the mentioned application example has not yet been developed. Therefore, no reference is available at the moment, and the probing of the spectral response of the Fabry-Perot interferometer using AFBS will be the topic of further research. The works dedicated to the concentration measurements of various gases using Fabry-Perot interferometers are cited in the sub-section 5.5 of the current manuscript.

Reviewer 2 Report

This review reported an comprehensive review over the fiber Bragg grating, including the structures with three or more spectral components with various combinations of difference frequencies, both symmetrical and asymmetric. I think this article should consider the following suggestions and revise them carefully before any consideration:   

1. As a review the reported article comprises very less literature and cited few only, I strongly suggest author to provide a comprehensive discussion over the history, background, need of the study, future challenges, outcome, etc. Authors are advised to follow uniform referencing.

2.  An graphical abstract or summary of work in figure form must be included in the article to gain more visibility.

3. The article is lack of schematic diagram and figures, and hence makes it less interesting. I personally suggest authors to add more figure and diagram.

4. Challenges and limitations are missing from each sections is missing from the article. A good review required an appropriate outlook (prospect).

5. Summary table must be included in each section and subsection to demonstrate all article reported so far in particular field.

Author Response

We would like to thank the Reviewer for the thorough review of the paper and the valuable remarks.

This review reported an comprehensive review over the fiber Bragg grating, including the structures with three or more spectral components with various combinations of difference frequencies, both symmetrical and asymmetric. I think this article should consider the following suggestions and revise them carefully before any consideration:   

  1. As a review the reported article comprises very less literature and cited few only, I strongly suggest author to provide a comprehensive discussion over the history, background, need of the study, future challenges, outcome, etc. Authors are advised to follow uniform referencing.

Thank you for the remark. As a result of the review, several references were added to the manuscript [20, 24, 37, 41, 52, 54]. As far as the authors are concerned, the current review now includes references to the most significant works dedicated to the topic of AFBS, which was initially introduced by the authors. The manuscript also includes comprehensive overview of the theoretical and technological aspects of AFBS implementation citing the most relevant works on the corresponding topics. Nevertheless, in case of necessity, the authors would be grateful if the Reviewer could suggest particular technological topics to be discussed or works to be cited in the current manuscript.

  1. An graphical abstract or summary of work in figure form must be included in the article to gain more visibility.

Thank you for the suggestion. A graphical abstract was added to the article.

  1. The article is lack of schematic diagram and figures, and hence makes it less interesting. I personally suggest authors to add more figure and diagram.

In order to make the review more vivid, we added Figure 4 illustrating the principle of AFBS interrogation, and Figure 13 depicting the spectral response of an ACFOS.

  1. Challenges and limitations are missing from each sections is missing from the article. A good review required an appropriate outlook (prospect).

Thank you for the suggestion. The paragraphs highlighting specific challenges and limitations of AFBS were added at the end of Section 3 Interrogation of Addressed Fiber Bragg Structures and Section 4 Fabrication Methods of Addressed Fiber Bragg Structures. Section 5 does not include separate paragraphs on this topic, since the challenges and limitations of AFBS usage in particular applications are defined by the previously mentioned aspects, and their repetition would be redundant.

  1. Summary table must be included in each section and subsection to demonstrate all article reported so far in particular field.

The summary tables listing the references were added at the end of Section 3, Section 4, and each of the subsections of Section 5.

Reviewer 3 Report

The manuscript entitled   “Overview of Addressed Fiber Bragg Structures’ Development” presents a good review on fiber Bragg gratings  based on the addressed frequency (AFBS structures). However, on my view, some modifications of this manuscript would be useful in order to it brings more information not only to experts in the field but also to readers interesting in sensors, generally. Thus the following changes are proposed to be carried out:

1.       It is not explained why are reflection spectra presented in Fig. 2 if such structures are interrogated in transmission set ups (Fig. 3 (b)) ?

2.       In Part 3, explain please, what is shown in two   output spectra b and c in Fig. 3?  Moreover, it is necessary to explain why are linear frequency responses of filters 3.1. and 3.2 necessary  and why should  have these filters different temperature sensitivities?

3.       It is not clear why is used a third power polynomial  in Eq. (5) and second power one in Eq. (6)? Moreover, by expressing the coefficients in Eq. 960 by second order polynomials is not explained. Thus, a reader would expect information in which papers such assumptions have been used, if they were evaluated statistically, etc.

4.       In Part 4.1., first paragraph, it is not clear if phase masks and a special mask [21] are used together as  the last sentence of the paragraph indicates. Furthermore, who establishes the technique described in second paragraph 4.1.? Has this technique been published?

5.       In Part 5.1. it  is necessary to explain the principle of RH measurements in more details, e.g., how are changes of polyimide coatings monitored with AFBS1. What is determined by means of AFBS2? An example of a real AFBS spectra and its changes would be helpful.

6.       In Part 5.5 please add what are properties of the detection films changed by detected gases? Does this sensor need a spectrometer for the registration of the light interference?  In such a case, one advantage of AFBS is not employed.

7.       Keywords should be changed in order to show that the manuscript deals with a review on addressed fiber Bragg gratings and that it presents their interrogation, fabrication, modeling and employment in physical sensors.

I recommend to distinguish two parts of Fig. 3  as (A) and (B) instead of (a) and (b). A modified manuscript will need a careful reading in order to avoid typing errors such as Wavelenght instead of Wavelength in Figs. 1 and 2.    

Author Response

We would like to thank the Reviewer for the comprehensive review of the paper and the valuable remarks.

The manuscript entitled   “Overview of Addressed Fiber Bragg Structures’ Development” presents a good review on fiber Bragg gratings  based on the addressed frequency (AFBS structures). However, on my view, some modifications of this manuscript would be useful in order to it brings more information not only to experts in the field but also to readers interesting in sensors, generally. Thus the following changes are proposed to be carried out:

  1. It is not explained why are reflection spectra presented in Fig. 2 if such structures are interrogated in transmission set ups (Fig. 3 (b)) ?

Thank you for the remark. The reflection spectra were replaced with the transmittance spectra for the Np-FBG type AFBSs (Fig. 1) in accordance with their interrogation approach.

  1. In Part 3, explain please, what is shown in two   output spectra b and c in Fig. 3?  Moreover, it is necessary to explain why are linear frequency responses of filters 3.1. and 3.2 necessary  and why should  have these filters different temperature sensitivities?

The diagram (b) is used to illustrate that the spectrum of the light source (green curve) covers all of the AFBS spectral components’ positions, while the diagram (c) shows the spectrum of the actual resulting output radiation from the AFBSs.

Regarding the linear frequency responses of filters 3.1. and 3.2, as we mentioned in the text (lines 149 – 151), “the deviation from the linear approximation of the optical filter frequency response is one of the main components of the measurement error, since the position of the AFBS spectrum is determined relative to it”. The reason behind this is that the amplitudes of the AFBS spectral components Ai and Bi used for the calculation of the AFBS spectral position relative to the filters 3.1 or 3.2 are defined by the parameters u (the slope) and v (the intercept) of the linear function describing the inclined frequency response of the filter (3.1) or (3.2) (equation (3)). However, it is also possible to use nonlinear function to describe the frequency response of the filters, although it would result in more complicated calculations of the AFBS spectral position relative to the filter.

Regarding the temperature sensitivities of the filters, it was mentioned in the text (lines 163 – 169), that “… in order to correctly determine the AFBS central wavelength, it is necessary to take into account the temperature drifts of the optical filters 3.1 and 3.2. For this reason, the filters are located close to each other in the system layout so that their temperature is assumed to be the same. Therefore, knowing the difference between the center wavelengths of the same AFBS determined using the filters, it is possible to calculate their temperature using the pre-defined temperature characteristics of the filters [17]”. After that, the estimated value of temperature is used to calculate the absolute value of the center frequency of the filter, based on which the correction to the center frequency of the AFBS is determined. The latter statement has been added to the text for clarification (lines 169 – 171).

  1. It is not clear why is used a third power polynomial  in Eq. (5) and second power one in Eq. (6)? Moreover, by expressing the coefficients in Eq. 960 by second order polynomials is not explained. Thus, a reader would expect information in which papers such assumptions have been used, if they were evaluated statistically, etc.

Thank you for the concern. The authors added citation of their original work [20], in which this calibration method was discussed. Prior to the publication, the authors had performed numerous experiments, which resulted in the usage of the third power polynomial for the strain sensing calibration, especially at wide ranges of strain variation. The polynomials of the fourth and higher order were also considered, however they had negligibly low coefficients at lower powers of the polynomial. Therefore, the authors proposed the usage of the third-order polynomial for strain sensing calibration in general case. However, in many specific cases, the strain sensing calibration curve can be described by the polynomial of lower order and even by linear relation. An example of work dedicated to the temperature sensor calibration is [Chen, Changhao & Wu, Qi & Xiong, Ke & Hongzhou, Zhai & Yoshikawa, Nobuhiro & Wang, Rong. (2020). Hybrid Temperature and Stress Monitoring of Woven Fabric Thermoplastic Composite Using Fiber Bragg Grating Based Sensing Technique. Sensors. 20. 3081. 10.3390/s20113081.], where the third-order polynomial was used for FBG temperature calibration. However, as it can be seen from the results of the mentioned work, the third coefficient in the resulting polynomial is very small (~10-10) and therefore can be omitted.

  1. In Part 4.1., first paragraph, it is not clear if phase masks and a special mask [21] are used together as  the last sentence of the paragraph indicates. Furthermore, who establishes the technique described in second paragraph 4.1.? Has this technique been published?

In [22, or 21 before revisions], only the special phase mask with a thickness difference was used. The works [21, or 20 before revisions] and [23, or 22 before revisions] use only conventional phase masks. The sentence was reformulated for clarification: “The works [21] and [23] use conventional phase masks to record FBGs, while a special mask with a thickness difference of 2300 nm was utilized in [22], using which a phase shift is formed at the place of the mask thickness step”.

The technique described in second paragraph 4.1. was proposed by the authors of the current manuscript, the citation of the original work was added.

  1. In Part 5.1. it  is necessary to explain the principle of RH measurements in more details, e.g., how are changes of polyimide coatings monitored with AFBS1. What is determined by means of AFBS2? An example of a real AFBS spectra and its changes would be helpful.

Thank you for the suggestion. The following was added to the text: “As it was shown in [34], the Bragg wavelength of the AFBS1 with polyimide coating linearly increases with the increase of the RH at constant temperature, due to the strain effect caused by the expansion of the polyimide when it absorbs the moisture.” “The AFBS2 is sensitive to the refractive index of the environment due to the etched cladding. The increase of the environmental refractive index causes the increase of the Bragg wavelength of the AFBS2 [36], thereby, the sensor can detect the condensed moisture.”

  1. In Part 5.5 please add what are properties of the detection films changed by detected gases? Does this sensor need a spectrometer for the registration of the light interference?  In such a case, one advantage of AFBS is not employed.

We thank the Reviewer for the remark. As it is mentioned in the text (lines 496 – 497), “the film is made of a polymer, the refractive index of which reversibly changes depending on the gas concentration”. At the same time, the environmental temperature also affects the spectral response of the Fabry-Perot resonator via its thermo-optic and thermal expansion coefficients. Indeed, the original article [49] published by the authors proposed the usage of Karunen-Loeff transform [56] as an option to separate the FBG and the FPR spectra for their interrogation, which requires the usage of a spectrometer. However, the authors believe that the interrogation scheme of CFOS proposed in [49] will ultimately allow to define the spectral positions of both the AFBS and the FPR, if the FPR is used as an optical filter with inclined spectral response, similarly to the filters 3.1. and 3.2 in Fig. 3. At the moment the authors are developing the theoretical and implementational aspects of this approach, as it requires the careful choice of the FPR material properties, its geometrical dimensions, etc.

  1. Keywords should be changed in order to show that the manuscript deals with a review on addressed fiber Bragg gratings and that it presents their interrogation, fabrication, modeling and employment in physical sensors.

Thank you for the suggestion. The keywords have been changed.

I recommend to distinguish two parts of Fig. 3  as (A) and (B) instead of (a) and (b). A modified manuscript will need a careful reading in order to avoid typing errors such as Wavelenght instead of Wavelength in Figs. 1 and 2.

Thank you for the careful review. The denotations of Fig. 3 were changed, and the typing errors were corrected.

Round 2

Reviewer 2 Report

From author's response it looks like they didn't considered the comments seriously as the responses are not satisfactory. Hence, I'm rejecting the manuscript and not willing to review it anymore. The comments are attached below:

1. My concern 1: article still do not provide a comprehensive review. Even after my comment author only cited 60 articles in the revised manuscript which shows the negligence of the previously published article in the field.

2. My concern 2: I asked authors to provide graphical abstract/summary, now I can see author responded the comment however didn't include one in the revised manuscript.

3. My concern 4: I asked author to include challenges and limitation while I can see section 5 where author shows several applications where they can discuss the advantage and limitation of AFBS but skipped with some excuses which is not acceptable. Each application has some limitations which are overcome by their alternative which should be included here for each application.

4. My concern 5: there are several article published for these applications whereas author just cherry picked few.